# Tectonic Implication of the 2022 *M*_S_ 6.9 Earthquake in Menyuan, Qinghai, China: Analysis of Precise Earthquake Locations and InSAR

**DOI:** 10.3390/s23042128

**Published:** 2023-02-13

**Authors:** Xinxin Yin, Hongyu Zhai, Run Cai, Jiangtao Qiu, Xiaobo Zou

**Affiliations:** 1Institute of Geophysics, China Earthquake Administration, Beijing 100081, China; 2Gansu Earthquake Agency, Lanzhou 730046, China; 3Chengdu Surveying Geotechnical Research Institute Co., Ltd. of MCC, Chengdu 610063, China; 4The Second Crust Monitoring and Application Center, China Earthquake Administration, Xi’an 710054, China

**Keywords:** 2022 Menyuan earthquake, Lenglongling fault, Tuolaishan fault, precise earthquake location, coseismic deformation

## Abstract

Precise earthquake locations and InSAR (Interferometric Synthetic Aperture Radar) deformation observation are the major methods to understand the earthquake occurrence and disaster-causing process. This paper proposes a processing framework for analyzing strong earthquake mechanisms from one-dimensional velocity inversion to precise earthquake locations combined with InSAR deformation observation, and discusses earthquake-generating fault and dynamic mechanisms of tectonic deformation. We analyzed the Menyuan *M*s 6.9 earthquake in 2022 and discuss the historical seismic activities and corresponding stress adjustment processes in the research region. To analyze and study the seismogenic structure and mechanism of the earthquake, we investigated the spatial and temporal distribution characteristics of the Menyuan earthquake sequence and analyzed the InSAR coseismic deformation field. We obtained the precise locations of the main shock and aftershocks and the coseismic InSAR deformation field of the main shock. It was confirmed that the *M*s 6.9 earthquake was a shallow sinistral strike-slip earthquake, which led to the sequential activation of the Tuolaishan and Lenglongling faults. The main seismogenic fault of the mainshock was the northwestern end of the Lenglongling fault, and the earthquake rupture was segmented. It can be inferred that the earthquake was a stress-adjusted event triggered in the Qilian-Haiyuan tectonic belt caused by the northeasterly push of the Qinghai-Tibet Plateau. The risk of moderate to high earthquakes in the region remains high in the future, requiring enhanced seismic observations.

## 1. Introduction

After the occurrence of strong earthquakes, precise earthquake locations and surface deformation observation are common research methods for understanding the mechanism and process of earthquake occurrence [1]. Accurate source location is an important parameter for studying the internal structure of the earth, seismogenic structure, fault distribution, source geometry and other issues [2,3]. The one-dimensional velocity model is often used for seismic positioning, but the criteria and methods for selecting the model are different: the one-dimensional velocity model obtained by others is directly quoted or slightly adjusted [4,5,6], or inversion based on the existing velocity model to obtain a one-dimensional velocity model [7]. Due to the regional differences of geological structures, the one-dimensional velocity models in different regions are quite different, so it is necessary to conduct targeted case analysis on specific cases [8]. The accuracy of seismic positioning is highly dependent on the velocity model. In order to obtain a more accurate one-dimensional velocity model, the method proposed by Kissling et al. (1994) to invert the optimal one-dimensional velocity model is a classical method to obtain a reliable one-dimensional velocity model [9,10]. In order to obtain more accurate distribution pattern of earthquake swarms, at present, many earthquake location studies adopt relative location method, among which the double difference location method [11] is the most frequently used relative location method. This method makes a difference in the records of two seismic events that are very close to each other at the same station and adjusts the relative position between two seismic events by the traveling time difference. Because the same ray path from two seismic events to the same station is eliminated in the process of making a difference, it reduces the impact of the velocity model on the positioning results to a certain extent, which is the main reason why the double difference positioning method is widely used [12]. The research based on precise seismic location can only help us understand the occurrence process of the earthquake in the underground source area, while the surface deformation and landslide caused by the earthquake are difficult to fully monitor by the seismograph only. InSAR, or synthetic aperture radar interferometry, uses microwave (1 mm~1 m) phase difference generated by repeated observations of the surface using microwave synthetic aperture radar (SAR) image data to calculate the surface deformation, and the accuracy can reach the millimeter level [13,14,15]. At present, InSAR technology is at the forefront of scientific and technological innovation in deep space earth observation and is in a rapid development period. It plays an important role in supporting the monitoring of earthquake disasters and is widely used in the analysis of multiple strong earthquakes [16,17,18,19,20]. It is a new trend for us to fully understand the mechanism of earthquake occurrence and the process of disaster generation by combining heaven and earth observation and analyzing from the surface to the ground in an all-around way.

According to the China Earthquake Networks Center (CENC) report, an *M*_S_ 6.9 earthquake occurred in Menyuan County, Qinghai Province, China, at 01:13 on 8 January 2022; the initial location of the epicenter was 37.77° N, 101.26° E, and the focal depth was 10 km. This earthquake was the third *M*_S_ > 6 event after the two earthquakes *M*_S_ 6.4 in 1986 and *M*_S_ 6.4 in 2016. According to a field investigation, the surface rupture length caused by this earthquake exceeded 20 km, with a maximum 2.1 m sinistral dislocation. A total of 4052 houses were damaged and nine people were injured in several counties near the epicenter, and the tunnels and facilities of the Haomen-Junmachang section of the Lanzhou-Xinjiang high-speed railway were seriously damaged. Therefore, this earthquake has aroused widespread concern in academia and the public. According to the tomographic imaging results of the upper and middle crust in the source area, the Mengyuan *M*s 6.4 (2016) and *M*s 6.9 (2022) earthquake occurred in the zone of dramatic changes in P- and S-wave velocities, and this region also corresponds to the transition zone of high and low values of Poisson’s ratio and saturation [21,22]. In addition, the *M*s 5.2 aftershock that occurred on 12 January was close to the 2016 earthquake, which suggests that this seismic activity led to a full release of the energy accumulated by the Lenglongling rupture, and that a larger magnitude earthquake is unlikely to occur in the near future [23]. However, the joint analysis from the optimal sliding model, aftershock relocation results, Coulomb stress variation, and field observations, reveals that the accumulated strain energy of the Tuolaishan rupture has not been fully released, and the future seismic hazard in this region still requires high attention [24]. Our purpose is to take this earthquake as an example, and to explain the occurrence process of the Menyuan earthquake by combining seismic precise positioning and InSAR surface deformation observation methods, to fully understand this earthquake.

## 2. Historical Earthquake and Tectonic Background of Menyuan Earthquake

### 2.1. Historical Earthquake Information of Menyuan

Strong earthquakes occur frequently in Menyuan area, and the structural characteristics are complex. It is worth noting that the 1986 Menyuan earthquake was mainly a normal fault type, while the 2016 Menyuan earthquake was a thrust type [14,25,26]. Based on the analyzing of coseismic deformation, aftershock relocations, and geological data, Zhang et al. [27] found the 2016 event was a potentially delayed event following the 1986 event, which ruptured on the same southwest-dipping Minyue-Damaying fault, but at a deeper position. A bulging fault block formed between the main and secondary Lenglongling fault was lifted further up under the northwest, pushed by the Tibetan Plateau, finally generating the Menyuan 2016 thrust earthquake [28]. However, the most recent event appeared to be a strike-slip event, and the tectonic properties also appear to be different [29]. Combining different SAR displacement observations, the maximum horizontal and vertical displacements of the Menyuan earthquake were 1.9 and 0.6 m, while the magnitude of fault-parallel displacement on the south side of the fault was larger compared to the north side [30]. The three earthquakes were quite different, which shows the complex tectonic movement background of the northeastern margin of the Qinghai-Tibet Plateau where the earthquakes occurred.

We downloaded the earthquake catalog from 2009 to 2022 from the CENC (http://10.5.202.37:8080/JOPENSCat, (accessed on 18 January 2022)). The distribution of earthquake epicenters in Menyuan and surrounding areas is shown in Figure 1c. It can be seen from the figure that the earthquake events in recent years mainly developed in the Tuolaishan fault and the Lenglongling fault. Over time, the seismic activity transferred from the Lenglongling fault to the Tuolaishan fault. The Menyuan *M*s 6.6 mainshock occurred at the western boundary of the Lenglongling fault near the boundary with the Tuolaishan fault. Owing to the regional stress adjustment caused by the mainshock, the following two aftershocks with magnitudes greater than *M*s 5.0 occurred on the eastern edge of the Tuolaishan fault and in the middle of the Lenglongling fault.

To analyze and study the seismogenic structure and mechanism of the earthquake in detail, we adopted the VELEST location method [31] and double-difference location method (HypoDD) [11] to accurately locate the events in the earthquake sequence. InSAR technology was used to process the SAR images of ascending and descending orbits between 5 and 17 January. According to the precise location of the earthquake and InSAR results, we discuss the structural form of the seismogenic fault and the seismogenic mechanism of this earthquake.

### 2.2. Tectonic Background

As the forefront of the Qinghai-Tibet Plateau’s peripheral expansion caused by the collision and convergence of the Indian plate and Eurasian plate, a series of NW-trending thrust faults and folds have developed along the northeastern margin of the Qinghai-Tibet Plateau. The geological structure in the region is extremely complex, and strong earthquakes occur frequently (Figure 1). A geological survey, GPS observation, and focal mechanism research show that this region is experiencing strong NE-trending deformation and shortening [32,33,34]. Menyuan is located at the forefront of the northeastern margin of the Qinghai-Tibet Plateau, and it has become a sensitive area for tectonic activity and stress field changes, where the tectonic stress field is complex and the direction of the block movement is unstable [35].

The two *Ms* 6.4 earthquakes that occurred in the Menyuan area in 1986 and 2016 indicated that the possibility of a strike-slip earthquake on this Lenglongling fault is gradually increasing [36]. The 2022 Menyuan *M*s 6.6 mainshock occurred on the western boundary of the Lenglongling fault near the boundary with the Tuolaishan fault. The Lenglongling fault is an important sinistral strike-slip fault on the northeastern margin of the Qinghai-Tibet Plateau, with a general strike of about 300°, an NE dip angle of 50–60°, and a fault bandwidth of 1–3 km [37,38]. He et al [39] found that the early activity of the Lenglongling fault was mainly compressional thrust, but in the late Quaternary it changed to a sinistral strike-slip and normal fault. Since the Holocene (Q4), the fault activity was mainly horizontal movement, with an average slip rate of 3.35–4.62 mm a^−1^ and an average vertical slip rate of 0.38 mm a^−1^. In the early stage, the Tuolaishan fault zone (strike 280°) was also dominated by compressional thrust. Since the late Quaternary it has changed to be dominated by sinistral thrust. Moreover, the fault was continuously active in the Late Pleistocene and has been active in the Holocene (Q3 and Q4) [40,41].

## 3. Data and Methods

### 3.1. Absolute Location of the Earthquake Sequence

According to the CENC monitoring records, from 8 to 15 January 2022, a total of 2357 active events of the Menyuan earthquake were recorded. Among them, 1648 earthquake events were recorded by a single station, and 508 earthquake events were recorded by more than four stations; 13 earthquakes had *M*s ≧ 3.0, and two aftershocks exceeded *M*s 5.0, i.e., *M*s 5.1 at 2:09 on 8 January and *M*s 5.2 at 18:20 on 12 January 2022. To ensure the reliability of the earthquake’s epicenter, we selected the seismic events (508 times) recorded by more than four stations for analysis. First, the VELEST location method was used to relocate the downloaded seismic events and optimize the velocity model; 500 km was selected as the threshold for the epicenter distance range, and the screened results of the seismic phase travel-time are shown in Figure 2.

The accuracy of the seismic location mainly depends on the selection accuracy of the seismic phase arrival and the velocity model used for location inversion calculations. An accurate one-dimensional velocity model can provide more accurate source information. At the same time, the minimum one-dimensional velocity model is reliably used in seismic location, which can minimize the root mean square (RMS) of travel time residual [31]. Velest algorithm is a Fortran77 program, which is used to obtain a one-dimensional velocity structure model in seismic location and use it as a reference model to improve the accuracy of seismic location or apply it to seismic tomography. An initial velocity model is selected first, and the initial model is corrected by correcting the station and source parameters during the program operation [42,43]. The initial velocity model used in this study refers to the Gansu-Qinghai one-dimensional velocity model, the results of velocity imaging research in the Menyuan area, and the velocity model used by related scholars in the Menyuan area [22,44,45,46]. After several calculations, the initial velocity model that was used is shown in Figure 3. For absolute seismic location and velocity model optimization using the VELEST method, we selected 100 seismic events with more than 12 station records from 508 seismic events as the velocity model optimization data. The optimal velocity model was obtained after 80 iterations (Figure 3). After optimizing the velocity model, considering the depth error of the earthquake catalog from the CENC, we continued to use the VELEST method and chose the location-only mode to relocate the 508 seismic events, thus obtaining the absolute location of all of the events. As a result, the location error, especially the depth error, of the initial earthquake catalog was reduced.

### 3.2. Double-Difference Method for Precise Location

To further improve the location accuracy of seismic events, we continued to use the double-difference location method to perform precise location processing based on the relocated earthquake catalog. The double-difference location method (HypoDD) proposed by Waldhauser and Ellsworth in 2000 has been widely used in microseismic and natural earthquake locations [47,48,49]. This method considers the velocity changes caused by the spatial inhomogeneity of the crustal medium. Then, according to the absolute arrival information of P- and S-waves, the distance of event pairs is not limited, which reduces the error caused by the assumption of constant wave velocity in ordinary location methods and increases the seismic location accuracy [50].

We selected 50 km as the upper threshold of the earthquake event pairing distance in precision location using the double-difference method. If the distance of earthquake event pairing exceeded the threshold, it was not considered. In addition, the number of earthquake events paired with the same earthquake was set to be no more than 10, and we used a total of 64,236 P-wave phases and 55,034 S-wave phases for the double-difference method for precise seismic relocation. Considering the different degrees of errors introduced in the manual selection process of the initial seismic phases, the seismic phase weight of the P wave was set to 1.0, and the seismic phase weight of the S wave was set to 0.6 in the calculation process of the precise location in this study [22]. The precise location result obtained in this paper is shown in Figure 4.

In total, acceptable location results of 384 *M*s 0.5–6.9 earthquakes were obtained after precise seismic location using the double-difference method. The spatial location of the main earthquake (*M*s 6.9) was 37.767° N, 101.276° E, and the depth was 12.3 km. Its horizontal location was at the intersection of the Lenglongling fault and Tuolaishan fault. The earthquake location results of this paper are consistent with those of other researchers [51,52,53]. The depth of the *M*s 5.1 aftershock at 2:9 on 8 January was 13.3 km, which indicated that the fault slip continued to extend and develop at deeper depths after the mainshock. After relocation, the earthquake distribution was more concentrated compared with the origin catalog, and the *M*s 5.2 aftershock on 12 January occurred on the Lenglongling fault. This indicates that Lenglongling fault was activated after certain stress accumulation. A comparison of the focal mechanism inversion results of major earthquakes (Table 1) indicated that the fault strike (290°) corresponding to the *M*s 6.9 main shock was between the fault strikes corresponding to the two aftershocks with magnitudes higher than *M*s 5.0 (*M*s 5.1: 264°, *M*s 5.2: 301°). The corresponding spatial distribution characteristics of the aftershocks that occurred near the Lenglongling Fault (LLLF) and the Tuolaishan Fault (TLSF) also verified the inversion results of the focal mechanism of the *M*s 5.1 and *M*s 5.2 aftershocks.

The BB’ profile near the mainshock was fitted with an 80° dip angle of the fault plane, while the fitted fault plane dip angle of the DD’ profile of the *M*s 5.2 aftershock on the Lenglongling fault was about 75°. Under the same tectonic stress, the two fault zones had similar fault tendencies. It is worth mentioning that the AA’ profile was about 4 km away from the surface of the Tuolaishan Fault in horizontal projection. Regardless, according to the fault dip analysis, Tuolaishan Fault may have a specific inclination in this section. Hence, the seismic activity in this section mainly occurred within the main fault of Tuolaishan.

### 3.3. InSAR Coseismic Deformation

InSAR technology uses the reflection of microwaves emitted by a satellite on the ground to calculate the phase difference, and then performs an inverse calculation to obtain the surface micro-deformation [13]. After the Menyuan earthquake, we downloaded the C-band Sentinel-1 SAR image data (interference width mode, IW) of two tracks covering the whole earthquake area. The specific data pairing information is shown in Table 2. We used GAMMA software [54] when performing the D-InSAR two-track method to process the Sentinel-1 data. Then, a multi-view ratio of 10:2 (range direction: azimuth direction) was used to perform image registration to generate differential interference images. During the de-leveling process, the required external elevation model used SRTM data at a 30 m resolution (the Shuttle Radar Topography Mission, https://dwtkns.com/srtm30m/ (accessed on 18 January 2022)). After the phase unwrapping of the phase differential interference results by the branch-cut method [55], an additional iterative step was used to correct the unwrapping error in the interferogram [56]. For the phase delay caused by atmospheric water vapor, an atmospheric phase delay model was established based on the existing digital elevation model and was removed from the original interferogram by using a program with Gamma. Finally, the geocoding output was performed, and the coseismic deformation field of the ascending/descending orbit of the Menyuan earthquake was obtained (Figure 5a,b). Based on the coseismic deformation field, we used the intensity tracking algorithm [57] to estimate the range and azimuth offset fields of the ascending/descending orbit interferogram. Then, with the use of the migration field and the coseismic deformation field, the 3D deformation field of the Menyuan earthquake was further calculated (Figure 5c–e). There should be a minimum of three measurements from different directions in order to measure 3D deformation. Ideally the LOS from the three positions should be close to orthogonal to get the best accuracy and reconstruction of the deformation field. Here, we used the range and azimuth offset fields and the ascending/descending InSAR deformation fields for 3D decomposition. More observations could further improve the deformation solution. The 3D deformation solution was obtained using least-squares estimation, which was constructed as an over-determined set of linear equations of the form (A)(x) = b for each point in the scene. There are M rows in A, with each row representing a single measurement unit magnitude look vector in ENU coordinates. The three elements of the (x) column vector are the deformation vector components in east, north and up, and the M elements in the column vector (b) are the deformation obtained from offset fields and the InSAR deformation fields.

Based on this earthquake’s InSAR coseismic deformation field, the two interferograms from the ascending and descending orbits indicated relatively obvious deformation information (Figure 5a,b), consistent with the results of other researchers [24,29]. The maximum rise and subsidence of the coseismic deformation of the ascending/descending orbits were about 59 cm and 78 cm, respectively, which showed opposite trends on both sides of the Tuolaishan fault and Lenglongling fault. The coseismic deformation of the ascending/descending orbit showed the opposite results, indicating that the surface deformation caused by the earthquake was dominated by horizontal deformation, which corresponds to a strike-slip type earthquake event. The earthquake caused a rupture length of about 10 km along Tuolaishan fault, and that of Lenglongling fault approached 20 km. This result is consistent with the focal mechanism solution of the mainshock in Table 1 and the surface rupture length of the field investigation mentioned above. Based on the spatial distribution characteristics of the three-dimensional components of the coseismic deformation field, the earthquake deformation was dominated by east-west strike-slip, with thrust components. The east-west deformation field showed that the maximum deformation occurred at the junction of the Lenglongling fault and Tuolaishan fault (Figure 5c). Thrust deformation and uplift deformation mainly occurred on the southwestern plate of the Lenglongling fault, of which both showed positive values (Figure 5d,e).

### 3.4. Inversion of Slip Distribution

There is a linear relationship between the slip (strike-slip and dip-slip components) on the fault plane and surface deformation. According to the InSAR deformation field (Figure 5), we further analyzed the rupture and slip characteristics of the fault plane of this earthquake. According to the spatial distribution of InSAR coseismic deformation, we set the fault slip model as 82 km along the strike and 20 km along the dip and divided the fault plane into 820 patches according to 2 km ×1 km. To improve the inversion efficiency, we down-sampled the 3D deformation field (Figure 5c–e) using the uniform sampling method, with sampling intervals of 500 m and 2000 m inside and outside the seismic deformation area, respectively, to ensure the deformation characteristics and sampling density of the earthquake area. Then, we used the SDM package [58] and Okada dislocation model to invert the slip distribution of the seismogenic fault plane using the parameters of the seismogenic fault determined in Table 1 as the initial reference values. In the inversion process, for the sliding amount of adjacent patches, the stress drop smoothing constraint was applied because a constant stress drop is a more realistic approximation from a physical perspective.

The slip distribution obtained by the inversion is shown in Figure 6. The coseismic rupture length of this Menyuan earthquake was about 34 km, and the slip distribution was mainly concentrated in the depth range of 0–13 km downward along the dip, which is a typical shallow tectonic earthquake. The fault rupture caused by this earthquake was mainly left-lateral strike-slip, with a maximum slip of 4.5 m at a depth of 6.5 km. The eastern part of the rupture showed a slight thrust component with a maximum slip of 1.2 m. The moment magnitude estimated from the slip distribution was *M*_W_ 6.55–6.78, which is similar to the magnitude in Table 1.

## 4. Discussion

### 4.1. Seismogenic Fault

The surrounding terrain of the Menyuan earthquake is complex, and the altitude varies greatly, which corresponds to the complex tectonic stress field in the Menyuan area and the unstable direction of block movement. The earthquake occurred at the intersection of Lenglongling fault and Tuolaishan fault, both of which have sinistral strike-slip characteristics. Our precise earthquake location results showed that the Menyuan earthquake sequence was distributed in the range of about 50 km from east to west and about 30 km from north to south. The main earthquake of *M*s 6.9 occurred at the intersection of the Lenglongling fault and Tuolaishan fault, the *M*s 5.1 aftershock that occurred on the same day was located to the west of the Tuolaishan fault, and the *M*s 5.2 aftershock that occurred on 12 January was located on Lenglongling fault to the east of the main earthquake. The InSAR coseismic deformation showed a butterfly-like deformation pattern. The similar apparent deformation size and opposite amplitude polarity characteristics of the two walls of the fault also indicated that the earthquake activity was mainly strike-slip.

Based on the precise location and InSAR results of the earthquake, we believe that the main seismogenic fault of the Menyuan *M*_S_ 6.9 earthquake is Lenglongling fault. The earthquake caused the SW hanging wall of Lenglongling fault to rupture westward toward Tolaishan fault and the NE footwall to rupture northwest toward the end of Sunan-Qilian fault. The rupture length of Tuolaishan fault caused by the earthquake was about 10 km, and that of Lenglongling fault approached 20 km. The slip distribution of this earthquake on the fault plane was mainly concentrated in the depth range of 0–13 km downward along the dip, with a maximum slip of 4.5 m. After the main shock, the aftershocks moved together in Tolaishan fault and Lenglongling fault.

### 4.2. Seismogenic Mechanism

This earthquake was the third strongest earthquake in Menyuan after the 1986 and 2016 *M*_S_ 6.4 earthquakes. However, unlike the 1986 event, which was a normal fault earthquake, and the 2016 event, which was a thrust earthquake, this event was a strike-slip earthquake. These findings indicate the complexity of the seismogenic environment in this region. The present crustal motion field shows that the Qilian-Haiyuan tectonic belt where Menyuan is located is experiencing continuous and strong lateral escape, south-north crustal shortening and strong vertical uplift due to the northeasterly push of the Qinghai-Tibet block and the Alashan block in the northeast, which indicates that the dynamic background of this area is mainly crustal shortening and sinistral shearing.

Based on this, we believe that the occurrence of this main shock was the result of a sinistral strike-slip dislocation at the intersection of the Lenglongling fault and the Tuolaishan fault when the Qinghai-Tibet block was pushed in a northeasterly direction, resulting in a critical state. The undischarged energy triggered the *M*s 5.1 earthquake on the Tuolaishan fault tens of minutes after the mainshock. The occurrence of these two strong earthquakes and other aftershocks released the stress accumulation state on the western side of the Lenglongling fault zone to a certain extent, but also caused the imbalance of the stress state on the eastern and western sides of the Lenglongling fault zone, and finally contributed to the occurrence of the *M*s 5.2 earthquake on 12 January. The CC’ profile in Figure 5b shows that the evolutionary trajectory of aftershocks advancing from west to east supports the reliability of the above view. In addition, based on the AA’ and CC’ profiles, the seismic events at the intersection of the two tectonic fault zones were deeply distributed and gradually became shallower when extending to both sides, which also outlines the characteristics of the gradual change of the fault activation depth in the two corresponding fault zones. Therefore, we infer that this earthquake was mainly a stress-adjusted event triggered in the Qilian-Haiyuan tectonic belt, which was caused by the northeasterly pushing of the Qinghai-Tibet Plateau.

### 4.3. Seismic Activity of Lenglongling Fault

The Qilian-Haiyuan tectonic belt, where Lenglongling fault is located, is an important boundary fault for adjusting tectonic stress along the northeastern margin of the Qinghai-Tibet Plateau, where there have been many strong earthquakes in the past one hundred years, including the Gulang *M*_S_ 8.0 earthquake in 1927, Minqin *M*_S_ 7.0 earthquake in 1954, Menyuan *M*_S_ 6.4 earthquake in 1986, Menyuan *M*_S_ 6.4 earthquake in 2016 and this *M*_S_ 6.9 earthquake. Among them, both the 1986 Menyuan earthquake and the 2016 Menyuan earthquake occurred at the two end of branch fault on the northern side of Lenglongling fault [37,59,60]. The large number of millennial recurrent active faults in the Lenglongling-Gulang integral fault region and the much shorter earthquake recurrence cycle than the average level suggested that earthquake risk may be greater if fault depth-segments are activated [27,61]. These findings show that the Lenglongling fault plays a key role in adjusting the huge NE-trending compressive stress in the Qilian-Haiyuan fault zone, which also results in Lenglongling fault as the area with the strongest NE extension and the most intense tectonic transformation on the Qinghai-Tibet Plateau.

It is well known that the occurrence of moderate and strong earthquakes is caused by the overall unstable sliding of a fault, and a decoupling layer is often required to provide the overall dislocation conditions for the fault during the unstable sliding process [62,63]. The magnetotelluric detection results showed that a southwestward low-resistance body began to appear at a depth of 5 km below the Lenglongling fault and extended downward, which indicates that there may be an obvious weak zone of mechanical strength beneath Lenglongling fault [64]. During the process of obvious sinistral shear, south-north crustal shortening and strong vertical uplift, the existence of this zone with weak mechanical strength promotes seismic creep, slip, and occurrence on the Lenglongling fault, which may be another reason for the high seismic activity along the Lenglongling fault. This seismic sequence has resulted in the release of accumulated energy from the Lenglongling fault and the Tuolaishan fault to varying degrees. However, with the continued north-eastward extrusion of the Qinghai-Tibet Plateau and the obvious physical differences between the north and south sides of the Lenglongling Fault, this fault zone will remain a high-risk area for strong earthquakes in the future [21,24,65].

## 5. Conclusions

To analyze the seismogenic process and structural characteristics of the Qinghai Menyuan *M*s 6.9 earthquake and its aftershock sequence, we analyzed the precise locations of the Menyuan earthquake sequence and InSAR coseismic deformation field and finally drew the following conclusions:(1)According to the relocations of the earthquake sequence, the *M*s 6.9 main earthquake (37.767° N, 101.276° E, and the depth: 12.3 km) occurred at the intersection of the Lenglongling fault and the adjacent Tuolaishan fault, while the following two *M*s 5.1 and *M*s 5.2 aftershocks occurred in the Tuolaishan and Lenglongling fault zone, respectively.(2)The focal mechanism solution of the earthquake and the spatial geometric distribution of the aftershock sequence exhibited prominent east-west distribution characteristics. The energy released by the *M*s 6.9 mainshock changed the local stress field and activated the Tuolaishan fault zone. Subsequently, many aftershocks gradually transitioned from the Tuolaishan fault to the Lenglongling fault. The *M*s 5.2 aftershock on 12 January was triggered during the eastward transmission of the stress effect.(3)The earthquake reflects the calibration of accumulated tectonic stresses inside the Qilian-Haiyuan tectonic belt, where the Lenglongling fault lies under the action of northeasterly compressive stress on the northeastern margin of the Qinghai-Tibet Plateau. The higher recurrence rate of moderate-to-high earthquakes in this region also indicates that the future seismic hazard remains high, and earthquake observation and early warning in this region should be strengthened.

## Figures and Tables

**Figure 1 sensors-23-02128-f001:**
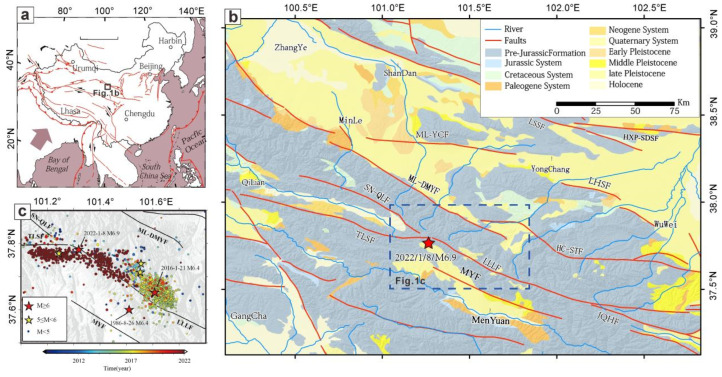
Regional tectonic map and historical earthquakes in the Menyuan region. The red star denotes the epicenter of the *M*s 6.9 Menyuan earthquake. LLLF: Lenglongling fault; TLSF: Tuolaishan fault; SN-QLF: Sunan-Qilian fault; MYF: Menyuan fault; ML-DMYF: Minle-Damaying fault. Information obtained from the earthquake catalog of the CENC. The 3 panels reflect the regional geological structure characteristics of this study from different spatial scales. Subfigure (**a**) shows the location of the Menyuan earthquake in China. Subfigure (**b**) shows the geological structure background of the Menyuan earthquake discussed in this paper. Subfigure (**c**) shows the spatial distribution of historical earthquakes in Menyuan from 2008 to 2022.

**Figure 2 sensors-23-02128-f002:**
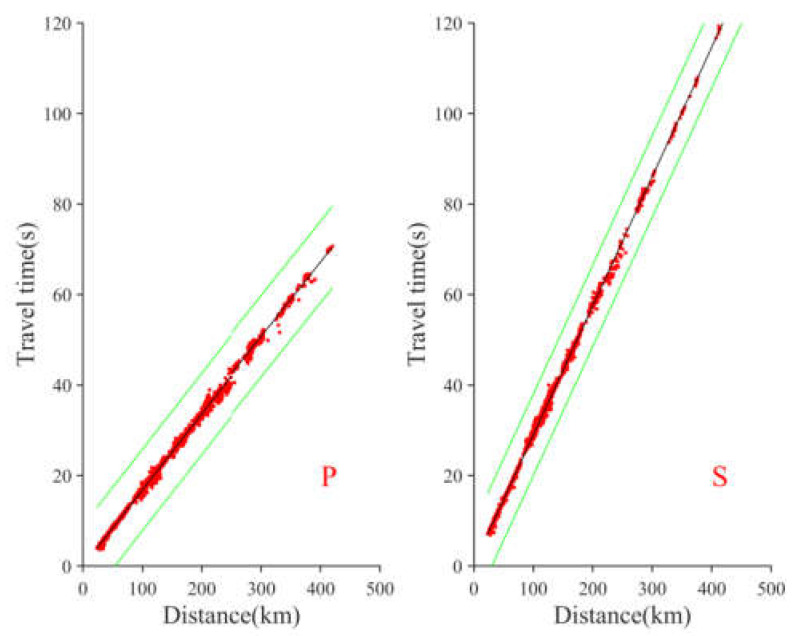
Observed (red) and predicted (black) travel times for the P (**left**) and S (**right**) phases. The green lines indicate the tolerance threshold (±9 s). Green line represents seismic phase screening boundary.

**Figure 3 sensors-23-02128-f003:**
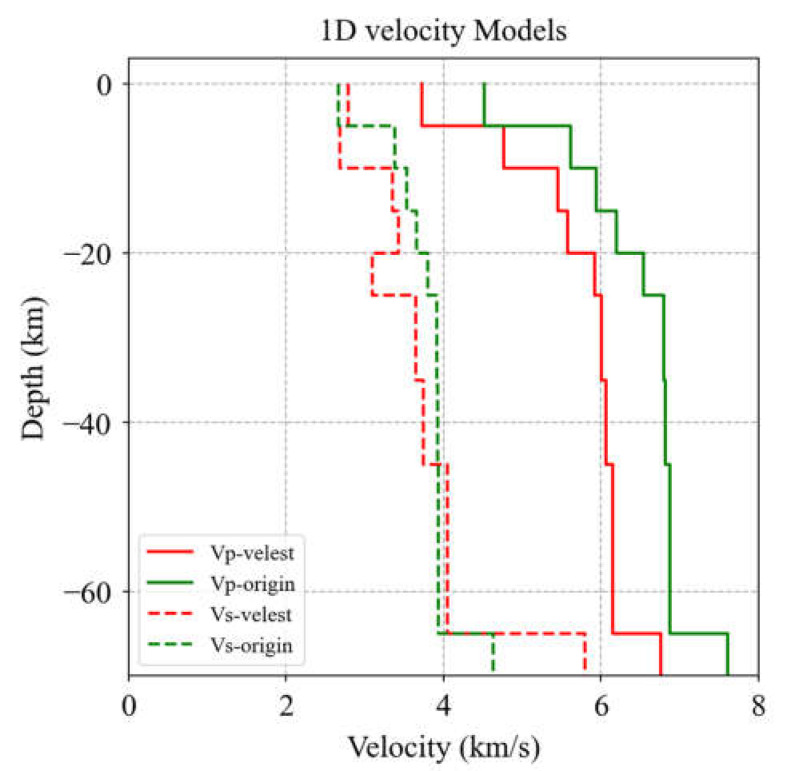
Initial 1-D velocity model (green line) and the optimized 1-D velocity model (red line) based on the VELEST method.

**Figure 4 sensors-23-02128-f004:**
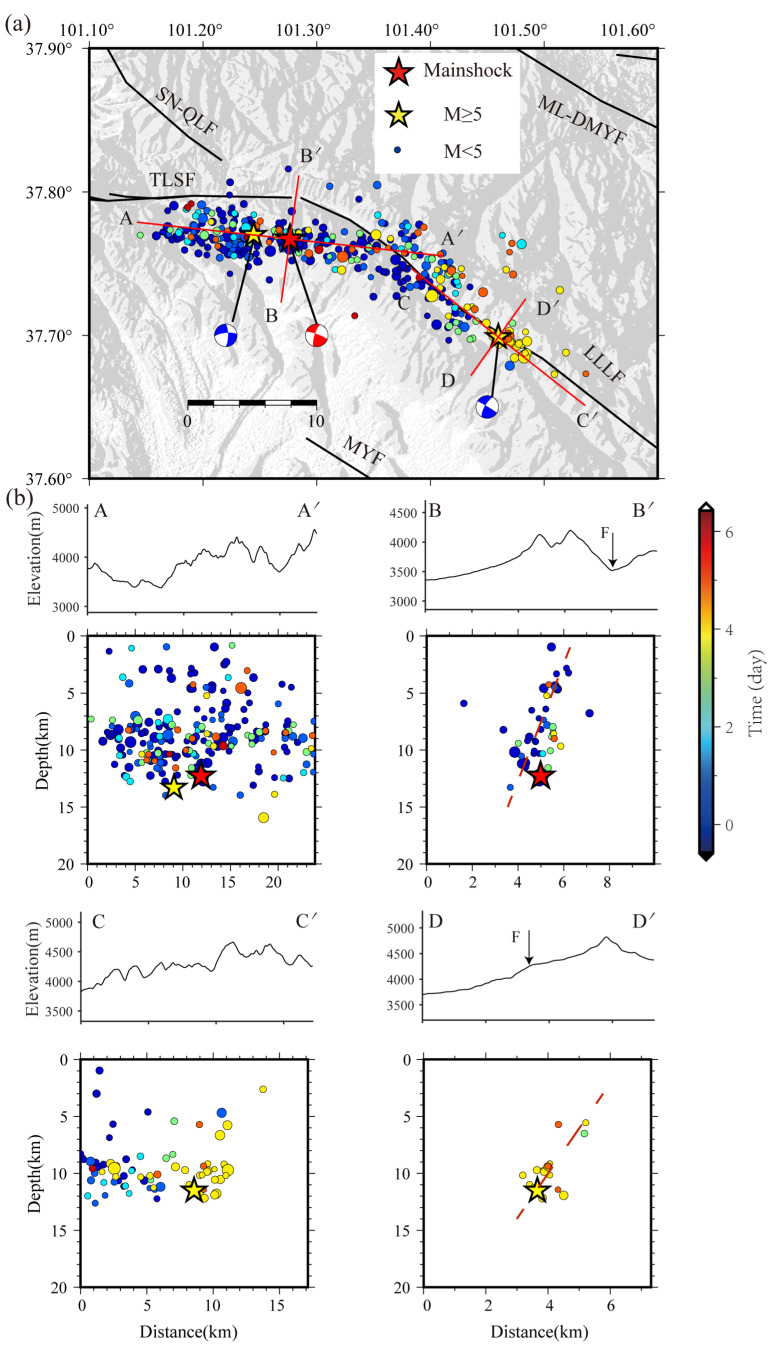
Map view (**a**) and depth distribution (**b**) of the aftershocks along four selected profiles. Earthquakes within 1.5 km of the selected profile lines are included.

**Figure 5 sensors-23-02128-f005:**
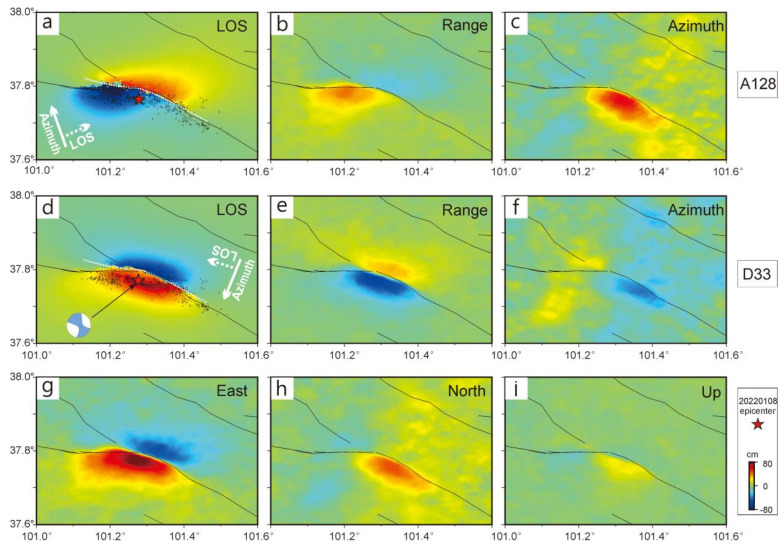
Coseismic deformation fields from InSAR data of the 2022 Menyuan *M*_S_ 6.9 earthquake; (**a**–**c**) are the InSAR deformation field and the range and azimuth offset fields of ascending track (A128); (**d**–**f**) are the InSAR deformation field and the range and azimuth offset fields of descending track (D33). Black circles are the earthquake locations after relocation. Solid arrows indicate the satellite flight direction, and open arrows are the radar look directions. (**g**–**i**) Three components of the coseismic deformation field in an east-west direction, south-north direction, and vertical direction; the deformations eastward, northward, and upward are positive.

**Figure 6 sensors-23-02128-f006:**
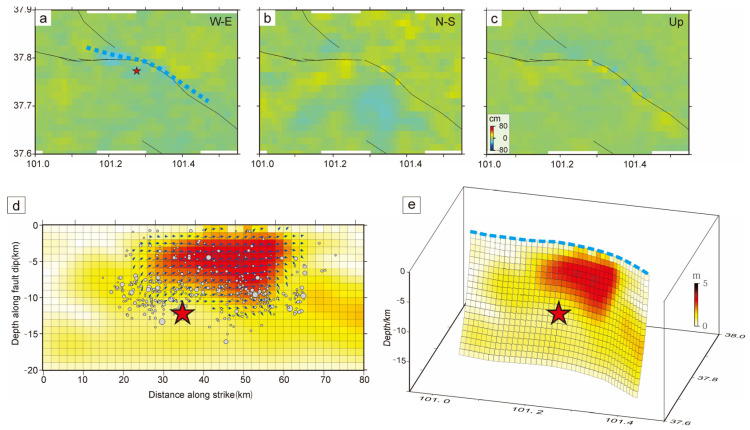
Inversion of the co-seismic slip distribution of the 2022 Menyuan earthquake. (**a**–**c**) Distribution maps of the residuals in the east-west direction, north-south direction, and vertical direction. (**d**,**e**) Two- and three-dimensional views of the co-seismic fault slip distribution of the earthquake.

**Table 1 sensors-23-02128-t001:** Focal mechanism solutions of the major earthquakes around Menyuan.

Time	Long. (°)	Lat. (°)	*M*s	Fault Plane 1			Fault Plane 2			Reference
				φ(°)	δ(°)	λ(°)	φ(°)	δ(°)	λ(°)	
21 January 2016	101.62	37.65	6.4	143	40	71	347	53	105	[46]
8 January 2022	101.26	37.77	6.6	290	81	16	197	74	171	[51]
8 January 2022	101.22	37.77	5.1	0	73	−161	264	72	−18	[51]
12 January 2022	101.47	37.70	5.2	210	72	177	301	87	18	[51]

**Table 2 sensors-23-02128-t002:** Parameters of SAR interferograms.

Model	Orbit No.	Image Date		Baseline Distance (m)	Time Interval (Day)
		Pre-Shock	AFTER-Shock		
ascending orbit	A128	5 January 2022	17 January 2022	38.4	12
descending orbit	D33	29 December 2021	10 January 2022	55.8	12

## Data Availability

The earthquake catalog used in this paper was provided by the China Earthquake Networks Center. Some data of this paper can be accessed from https://github.com/20041170036/Menyuan.git (accessed on 20 January 2023).

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
