# Peer review of "Tectonic Implication of the 2022 MS 6.9 Earthquake in Menyuan, Qinghai, China: Analysis of Precise Earthquake Locations and InSAR"

_sensors, 2023, doi:10.3390/s23042128_

Round 1

Reviewer 1 Report

-          Please explain the differences between the 3 panels of figure 1?

-          - Page 8, Lines 258-260, “ For the phase delay caused by atmospheric water vapor, an atmospheric phase delay model was established based on the existing digital elevation model and was removed from the original interferogram”

 Please explain in more details  how do you remove delay caused by atmospheric water vapor in phase difference?

-          Page 8, Lines 264, Please explain in more details “the intensity tracking algorithm”.

-          Page 9, Please Explain how do you calculate “(c–e) Three components of the coseismic deformation field in an east-west direction, south-north direction, and vertical direction;” ? 

 -          Figure 6 (a-c), Color-map is not clear. Cm or mm?

Author Response

We would like to thank you for your positive and constructive comments. These comments are all valuable and helpful for improving our article. Here are our Point-by-point responses to the reviewers.

Point 1: Please explain the differences between the 3 panels of figure 1?

Response 1: The 3 panels reflect the regional geological structure characteristics of this study from different spatial scales. Subfigure (a) shows the location of the Menyuan earthquake in China. Subfigure (b) shows the geological structure background of the Menyuan earthquake discussed in this paper. Subfigure (c) shows the spatial distribution of historical earthquakes in Menyuan from 2008 to 2022.

Point 2: Page 8, Lines 258-260, “ For the phase delay caused by atmospheric water vapor, an atmospheric phase delay model was established based on the existing digital elevation model and was removed from the original interferogram” . Please explain in more details how do you remove delay caused by atmospheric water vapor in phase difference?

Response 2: The atmospheric phase delay correction is carried out using the program atm_mod that comes with Gamma. The program evaluates linear regression of the unwrapped phase with respect to height and calculates a model of the atmospheric phase.

Point 3: Page 8, Lines 264, Please explain in more details “the intensity tracking algorithm”.

Response 3: We use the program offset_pwr_trackingm that comes with Gamma to estimate range and azimuth offset fields for detected (MLI) images using intensity tracking. We supplemented the reference (Magnard et al., 2017) in the manuscript, from which more details can be obtained.

Point 4:  Page 9, Please Explain how do you calculate “(c–e) Three components of the coseismic deformation field in an east-west direction, south-north direction, and vertical direction;” ? 

Response 4: We have supplemented some details about the 3D deformation solution in the manuscript.

Point 5:   Figure 6 (a-c), Color-map is not clear. Cm or mm?

Response 5: Thanks. We revised the color map by turning up the label in the color bar to improve clarity.

Reviewer 2 Report

I admire the authors due to they considered the performance of all the faults around the earthquake area and they had good discussion on the seismicity. I personally confirm that the structure of the article (except for very minor things like subtitles), the methods, and the data analyzes follow a good international standard for Sensors journal. So, I do not suggest a critical comment. 

Author Response

We would like to thank you for your positive  comments. Thanks very much. 

Reviewer 3 Report

This manuscript “Tectonic implication of the 2022 Ms 6.9 earthquake in Menyuan, Qinghai, China: Analysis of precise earthquake locations and InSAR” proposes a processing framework to obtain the precise locations of the mainshock and aftershocks, then uses the InSAR deformation observation to inverse the static slip distribution of 2022 Menyuan Ms 6.9 earthquake. This manuscript is clear and may attract some readers to understand the precise earthquake locations from a new processing framework and the details of the earthquake rupture process. The paper is generally well written and the subject is appropriate for Sensors. However, it requires some revision before ready for publication. I raise one major concern and some minor comments as below.

Major concern:

My main concern is that the manuscript lacks an introduction to previous studies on the 2022 Menyuan earthquake and a comparison of results. First of all, for the precise positioning of the main shock and aftershocks, I understand that the paper uses the Velest location method to obtain the one-dimensional velocity structure model, and then uses the double-difference location to perform precise location processing, which may help improve location precision. But there have been many previous studies on the precise location of the 2022 Menyuan earthquake, such as Xu et al. 2022 (Relocation and focal mechanism solutions of the Ms 6.9 Menyuan earthquake sequence on January 8, 2022 in Qinghai Province) and Yin et al. 2022 (Three-dimensional velocity structure and seismogenic mechanism of Menyuan Ms 6.9 earthquake in 2022), suggested a comparison with previous earthquake precise location results is added to the manuscript to illustrate the reliability of the method used in the manuscript. In addition, there have also been many research on the slip distribution inversion of the Menyuan earthquake, such as Lyu et al. 2022 (Joint inversion of InSAR and high-rate GNSS displacement waveforms for the rupture process of the 2022 Qinghai Menyuan M6.9 earthquake). It is recommended to add the comparation with previous studies to the manuscript.

Minor comments:

1. Line 79: The chapter label is wrong, and this section should be ‘2. Study of earthquake cases and tectonic background’.

2. Line 81: It is recommended to add this part to ‘1. Introduction’.

3. Line 206: Please explain why this weight is set in this way, or give relevant references.

4. Line 264: Please give references for the intensity tracking algorithm.

5. Line 267: Please give the azimuth offset in Figure 5.

Author Response

We would like to thank you for your positive and constructive comments. These comments are all valuable and helpful for improving our article. Here is our Point-by-point responses to the reviewers.

Point 1: My main concern is that the manuscript lacks an introduction to previous studies on the 2022 Menyuan earthquake and a comparison of results. First of all, for the precise positioning of the main shock and aftershocks, I understand that the paper uses the Velest location method to obtain the one-dimensional velocity structure model, and then uses the double-difference location to perform precise location processing, which may help improve location precision. But there have been many previous studies on the precise location of the 2022 Menyuan earthquake, such as Xu et al. 2022 (Relocation and focal mechanism solutions of the Ms 6.9 Menyuan earthquake sequence on January 8, 2022 in Qinghai Province) and Yin et al. 2022 (Three-dimensional velocity structure and seismogenic mechanism of Menyuan Ms 6.9 earthquake in 2022), suggested a comparison with previous earthquake precise location results is added to the manuscript to illustrate the reliability of the method used in the manuscript. In addition, there have also been many research on the slip distribution inversion of the Menyuan earthquake, such as Lyu et al. 2022 (Joint inversion of InSAR and high-rate GNSS displacement waveforms for the rupture process of the 2022 Qinghai Menyuan M 6.9 earthquake). It is recommended to add the comparation with previous studies to the manuscript.

Response 1:

We have added the latest literature related to the surface deformation and precise positioning of the 2022 Menyuan earthquake, compared the results, and found that the results of this paper are consistent with other results. In other studies, precise seismic positioning and surface InSAR deformation observation are rarely combined. This paper provides a comprehensive interpretation of the earthquake occurrence process from surface to underground for reference.

Point 2: Line 79: The chapter label is wrong, and this section should be ‘2. Study of earthquake cases and tectonic background’.

 Response 1: We revised the chapter label of the full text.

Point 2: Line 81: It is recommended to add this part to ‘1. Introduction’.

Response 2: We added the paragraph to the introduction.

Point 3: Line 206: Please explain why this weight is set in this way, or give relevant references.

Response 3: Because the pickup accuracy of the S-wave seismic phase is relatively low compared with that of the P-wave seismic phase, we set the weight with reference to previous results. And we have added relevant references (Zuo and chen, 2018) to the original text.

Point 4: Line 264: Please give references for the intensity tracking algorithm.

Response 4: We use the program offset_pwr_trackingm that comes with Gamma to estimate range and azimuth offset fields for detected (MLI) images using intensity tracking. We supplemented the reference (Magnard et al., 2017) in the manuscript, from which more details can be obtained.

Point 5: Line 267: Please give the azimuth offset in Figure 5.

Response 5: We added the azimuth offset in Figure 5.

Round 2

Reviewer 1 Report

After this revision this paper could be considered to be published.